# Defining Lyfe in the Universe: From Three Privileged Functions to Four Pillars

**DOI:** 10.3390/life10040042

**Published:** 2020-04-16

**Authors:** Stuart Bartlett, Michael L. Wong

**Affiliations:** 1Division of Geological and Planetary Sciences, California Institute of Technology, Pasadena, CA 91125, USA; 2Earth-Life Science Institute, Tokyo Institute of Technology, Tokyo 152-8550, Japan; 3Department of Astronomy and Astrobiology Program, University of Washington, Seattle, WA 98195, USA; miquai@uw.edu; 4NASA Nexus for Exoplanet System Science’s Virtual Planetary Laboratory, University of Washington, Seattle, WA 98195, USA

**Keywords:** definition of life, origin of life, astrobiology, mechanotroph

## Abstract

Motivated by the need to paint a more general picture of what life is—and could be—with respect to the rest of the phenomena of the universe, we propose a new vocabulary for astrobiological research. Lyfe is defined as any system that fulfills all four processes of the living state, namely: dissipation, autocatalysis, homeostasis, and learning. Life is defined as the instance of lyfe that we are familiar with on Earth, one that uses a specific organometallic molecular toolbox to record information about its environment and achieve dynamical order by dissipating certain planetary disequilibria. This new classification system allows the astrobiological community to more clearly define the questions that propel their research—e.g., whether they are developing a historical narrative to explain the origin of life (on Earth), or a universal narrative for the emergence of lyfe, or whether they are seeking signs of life specifically, or lyfe at large across the universe. While the concept of “life as we don’t know it” is not new, the four pillars of lyfe offer a novel perspective on the living state that is indifferent to the particular components that might produce it.

## 1. Introduction: The Need for a New Definition of Life

We contend that most standard definitions of life are restrictive and may blind future astrobiological research from life that is hiding in plain sight. NASA’s current working definition of life is “a self-sustaining chemical system capable of Darwinian evolution.” While this is a fair description of the life that is present at this moment on planet Earth, searching the universe for phenomena that fit this definition is similar to playing darts by focusing solely on the bullseye. For the uninitiated darts player, there are three problems with aiming only for the bullseye in a game of darts: (1) It is hard to hit because it occupies a tiny area of the dart board; (2) It is not the highest scoring region on the board (triple 60 for 180 is); (3) There are other high-scoring regions that should also be considered as targets. Hence, in the search for extraterrestrial life, we must consider that: (1) Life exactly as we know it may be rare in the universe, but a more general class of phenomena with life-like characteristics may be far more common; (2) There may be systems, yet to be discovered or even imagined, that more successfully satisfy the living criteria than even earthly life does; (3) By loosening our constraints on the definition of life, we open ourselves up to exploring the full parameter space of physical and chemical interactions that may create life.

We are also motivated by the often heated nature of the origins-of-life field, which has branched into various camps, each seeking to describe their own “one true” scenario for the origin of life. We posit that most of the contentious behavior within the field stems from differences in assumptions about what constitutes life and therefore what life’s onset must have been. As [1] wrote, “Without a definition for life, the problem of how life began is not well posed.” By redefining the vocabulary of the living state, we hope to engender greater mutual understanding and communication between the various communities engaged in origins-of-life research.

The remainder of Section 1 reviews some of the major issues leading to discord in the origins-of-life field that compelled us to seek a more general definition of life. In Section 2, we introduce our new expansive definition of life—which we call lyfe (pronounced “loif”)—and discuss how such a definition might engender new perspectives on origins-of-life research. Finally, in Section 3, we take the first steps toward imagining phenomena that could be classified as lyfe.

### 1.1. Privileged Functions at the Origin of Life

Many origins-of-life theories focus on explaining the emergence of a “privileged function”—a specific aspect of modern-day earthly biology that is assumed to have been present at its emergence. Often, these privileged functions are also implicitly assumed to be more fundamental to life in that once the privileged function is established, the rest of life’s functions should naturally emerge [2].

Another way of viewing privileged functions is the idea of conserved quantities, essential in physics and engineering. Origins researchers often try to pin down functional or material aspects of life that they believe must have been present and conserved from the outset. So important are such characteristics that without them, life no longer deserves to be called life (according to such an approach). One example is the so-called “chemistry conservation principle” as invoked by [3]. Their argument is summarized thus: “The chemical traits of organisms are more conservative than the changing environment and hence retain information about ancient environmental conditions” [4]. Under this principle, a search for the context of life’s origins is essentially a search for prebiotic environments that match cellular conditions as closely as possible.

Examples of privileged functions include template-driven replication of RNA, sets of reactions forming early metabolisms, and compartmentalization by lipid membranes (see Figure 1). This leads to the various “X-first” theories for the emergence of life. For instance, in the so-called “strong RNA-first” theory, “Darwinian evolution is thought to be necessary for nonintelligent matter to self-organize to produce behaviors that we ascribe to biology” [5], and the onset of Darwinism would occur with the abiotic formation of long-chained RNA polymers. Conversely, in many “metabolism-first” theories, life is viewed “like other self-organizing systems in the Universe, as an inevitable outcome of particular disequilibria” [6]. In this view, life, first and foremost, serves to dissipate these specific disequilibria, using disequilibria conversion engines to maintain its own low-entropy state [7,8]. Furthermore, in “compartment-first” theories, “compartmentalization of primitive biochemical reactions within membrane-bound water micro-droplets is considered an essential step in the origin of life” [9]. This spatial organization and molecular crowding is often mediated by self-assembling lipid vesicles, conferring heightened functionalilty to other biomolecules [10,11].

The focus on privileged functions leads these theories to different favored environments for life’s emergence—a source of great tension within the community. Laboratory experiments indicate that surficial environments—energized by UV radiation, exposed to (perhaps transient) reducing atmospheric conditions, and/or supplied with the relevant molecules from exogenous sources—are ideal for RNA synthesis, e.g., [12]. On the other hand, deep-sea alkaline hydrothermal vents appear to be the focusing centers for the redox and pH disequilibria that could drive the onset of proto-metabolic pathways, e.g., [6]. However, geothermal pools are favored for the spontaneous membrane assembly of lipid vesicles due, in part, to their low concentrations of divalent cations [13].

Every theory seeking to explain a privileged function of earthly biology contains the implicit assumption that the privileged function was present at the emergence of life and that such a function is fundamental to life. However, given the inherent lack of fossil or geological records, there is little to no evidence that any of these functions actually were present at life’s beginnings. Based on phylogenetic studies, our knowledge of Earth’s earliest lifeforms ceases at the Last Universal Common Ancestor (LUCA). Such studies have shed light on LUCA’s nature—most likely a cellular being with a chemiosmotic metabolism whose genetics were written in DNA/RNA. However, LUCA was almost certainly not the original lifeform on Earth (see, e.g., [14]). Furthermore, phylogenetic reconstructions of LUCA are inherently problematic and potentially unreliable due to the limits of bioinformatic techniques.

### 1.2. The Event Horizon in Origins-of-Life Research

There exists an “event horizon,” to appropriate a term from astrophysics, in origins-of-life studies (Figure 2). “Top-down” approaches, such as molecular phylogenetics, use clues in extant life to trace the history of life back towards its origin. These can take us back only as far as LUCA. “Bottom-up” approaches, which aim to simulate the synthesis of prebiotic molecules and/or the onset of proto-living structures and functions, may one day succeed in producing an abiogenesis. However, this “test-tube origin” will not be LUCA.

Furthermore, top-down approaches are limited in their insight into the origin of life because the tree of life that we see today can be satisfied by numerous theoretical histories (Figure 3). These histories can involve multiple geneses that never converge, resulting in completely different lineages that either go extinct or are still present as a “shadow biosphere” [15]. Plausible origins-of-life narratives can also involve horizontal gene transfer between different lineages, perhaps rampant in a protobiological ribofilm [16], such that LUCA is actually a plurality of lifeforms: Last Universal Common Ancestral Set (LUCAS) [17]. If so, then the “tree” of life may be a more fitting metaphor than initially imagined: while it branches upward into the diversity of beings that have inhabited our planet post-LUCA(S), so too does it radiate downward in a tangled root web of early living experiments to which we will never be privy.

### 1.3. Historical vs. Synthetic vs. Universal Origin Narratives

There is a deeper underlying issue: different origins-of-life researchers may not be seeking analogous explanations. Ref. [18] defined three distinct categories of origin narratives: historical, synthetic, and universal (Figure 4).

Historical origin narratives deal with the origin of life on Earth, constrained by our knowledge of early Earth environments and informed by what we observe in the products of that process: LUCA and the modern tree of life. All privileged function theories (Section 1.1) are historical narratives.

Synthetic origin narratives describe experiments where researchers direct the creation of new life in the laboratory. Such narratives may deal with the transition between nonlife and life, or they could create new forms of life from pre-existing forms of life, via directed evolution [20] or an artificially expanded basis set for encoding information [21]. As previously discussed, artificial abiogenesis experiments may be tuned to approximate historical narratives, but caution is warranted in conflating the two, due to the “clean lab” nature of artificial experiments and the vast uncertainty about prebiotic environments.

Universal narratives describe steps that are necessary for abiogenesis anywhere. They are relatively unconstrained by the details of the early Earth, the biosphere’s particular trajectory on our planet, or the chemical nature of life as we know it—all of these are but a single expression of the phenomena that result from a universal narrative. It is not yet certain that universal narratives exist, but some attempts have been made to explain life’s functions from an abstract, fundamental physics perspective, e.g., [22,23,24,25].

Hence, the different narratives that have been developed in origins-of-life research do not address a single scientific question but many. One commonly addressed question is the origin of life on Earth, which is naturally extended to the genesis of earthly life on other worlds. However, there is a completely separate question of the origin of life-like systems, whether earthly or not, in alien environments. Yet another question asks how we might create, in a completely synthetic fashion, new structures that we would deem living. All origins-of-life questions—be it to explain the origin of chemiosmotic coupling in earthly life, the circumvention of the “error catastrophe” in any template-driven replicating system, or the creation of dynamical order in general—are valid, but they are not necessarily asking the same thing.

These disparate aims and goals can cause the origins-of-life community to suffer from confusion (where there should be mutual understanding), friction (where there should be an amicable exchange of ideas), or apathy (where there should be genuine interest in one another’s progress). This paper introduces a new vocabulary and a new definition of life in response to these concerns.

## 2. The Definition of Lyfe

We seek to reframe the definition of life in a more expansive way while recognizing the need to signify the specific kind of life that earthly forms represent. Thus, we have come up with a new term—lyfe. Henceforth, we will refer to life (as we know it) and lyfe (as it could be, in the most general sense). The two designations are distinguished as follows:Life represents life as we know it; it uses the specific disequilibria and classes of components of earthly life. **Life is an autocatalytic network of organometallic chemicals in aqueous solution that records and processes information about its environment in molecular form and achieves dynamical order by dissipating any subset of the following disequilibria: redox gradients, chemiosmotic gradients, visible/thermal photons, etc.**Lyfe represents any hypothetical phenomenon in the universe that fulfills the fundamental processes of the living state, regardless of the disequilibria or components that it harnesses or uses. **Lyfe is any hypothetical phenomenon that maintains a low-entropy state via dissipation and disequilibria conversions, utilizes autocatalytic networks to achieve nonlinear growth and proliferation, employs homeostatic regulatory mechanisms to maintain stability and mitigate external perturbations, and acquires and processes functional information about its environment.**

The concept of “life as we don’t know it” is not new. However, traditional definitions of life struggle to clearly delineate boundaries between “life as we know it” and “life as we don’t know it.” Furthermore, many traditional definitions are equally fuzzy with regard to when “life as we don’t know it” becomes sufficiently far from “life as we know it” that it should no longer be considered life. For instance, according to NASA’s definition of life (Section 1), should a self-sustaining chemical system that is found evolving in a non-Darwinian fashion be considered “life as we don’t know it” or not alive at all?

To remedy this, we have developed our criteria for lyfe based on four fundamental processes. We agree with the general sentiment of [26] that “life is a verb, not a noun,” a remark based on the view that life operates by dissipating planetary redox gradients, shuttling electrons and transducing that disequilibrium into other dynamic configurations [27]. While the dissipation of free energy is certainly the first necessary aspect of life, we contend that it must be accompanied by three other processes—autocatalysis, homeostasis, and learning—to form a sufficient description of the living state.

In other words, we define lyfe as any physical system that exhibits all four of the processes described below. (Furthermore, life fulfills each of these via the specific mechanisms described in italics.)

**Dissipation**—Lyfe cannot exist at equilibrium. The second law of thermodynamics, in the presence of free energy transduction mechanisms, permits the coupling of exergonic processes to the endergonic, organized configurations of lyfe.
*Using an array of nanoscale molecular machines, life dissipates external chemical disequilibria and/or converts low-entropy photons into high-entropy waste heat, transducing these disequilibria into other disequilibria (e.g., endergonically building up proton gradients and high [ATP]/[ADP]). To perform useful work, life converts $ATP\to ADP+P_{i}$, which dissipates the [ATP]/[ADP] disequilibrium [28,29].*
**Autocatalysis**—The ability of a system to exhibit exponential growth of representative measures of size or population in ideal conditions. The property of autocatalysis can appear in different forms—including self-catalysis, cross-catalysis, and network autocatalysis—as long as the effect leads to exponential growth of a suitable metric under ideal conditions.
*A cultured system of microorganisms exhibits autocatalytic population growth due to cellular replication in resource-abundant conditions.*
**Homeostasis**—The ability of a system to maintain key internal variables within ranges of ideal set points. In a dynamic world of perturbations, coupled with the exponential growth described above, a lyving system must have means to limit the variation of its internal systems when external conditions change.
*Life performs homeostasis with networks of sensors, receptors, and effectors. The substance under homeostatic regulation (e.g., calcium ions) typically binds with receptors and promotes the release of further substances (e.g., hormones). These indicator compounds then stimulate an appropriate response mechanism to return the substance level to within the desired window.*
**Learning**—The ability of a system to record information about its external and internal environment, process that information, and carry out actions that feed back positively on its probability of surviving/proliferating.
*Darwinian evolution is one commonly cited biological learning process (e.g., [30,31,32]) among a much larger set of learning processes that living systems perform. For example, there are widely studied examples of biological learning within the realm of neuroscience, permitted by a range of neuronal and synaptic interactions (e.g., [33,34,35]). In addition, there is a growing list of non-neural learning systems, including gene regulatory networks [36,37,38], protein interaction networks [39,40], and other epigenetic mechanisms (e.g., [41,42]). Many examples fall under the general framework of associative learning, which has been exhibited by non-neural organisms such as slime moulds [43,44]. Darwinism mingles with these other learning processes (and perhaps other hitherto undiscovered forms) to create the incredible diversity and complexity of the biosphere. Hence, “learning” is an umbrella term for this large and incompletely understood set of processes.*


While these four pillars of lyfe are derived from observations of life as we know it (after all, life must be a subset of lyfe), this new definition is far more expansive. The four pillars constitute necessary and sufficient requirements of the lyving state while remaining divorced from the specific components that make up the system. This is illustrated by the fact that there are numerous systems that perform the same pillars but are quite distinct in form (as will be discussed in Section 2.1). Hence, the universality of the term “lyfe” is derived from the reasonable expectation that it can be applied to as yet undiscovered (or uninvented) systems that exist at myriad scales across the universe. There may even be a class of systems, still undiscovered and undescribed, that performs all four pillars of lyfe and some fifth pillar as well. Such systems might be deemed super-lyfe. While the discovery of super-lyfe would certainly be paradigm-shifting, we remain agnostic about its existence for now.

We take this moment to emphasize that our definition of lyfe applies at the system level. The pillars that can be ascribed to a certain system depend specifically on the boundaries with which we use to refer to that system. For example, which pillars do viruses perform? A single virus in isolation cannot perform any of the pillars. Viruses in a system consisting of viruses, bacteria, and nutrients can perform autocatalysis and, through the coercion of their bacterial hosts, dissipation. Viruses in a biosphere that is coevolving with its environment will cause not only autocatalysis and dissipation but learning (through evolution) as well. In certain ecosystems, viruses may even impart homeostatic attributes to the system by introducing auxiliary metabolic genes to their hosts and recycling organic matter via lysis [45]. Thus, the argument about whether a virus (or a mule, to cite another classic paradox) is alive becomes irrelevant under the assumption that “lyfeness” does not emerge at the molecular, cellular, or organismal level. Like others have suggested [46,47], we contend that the living state may best be assessed at an ecosystem or planetary scale.

As an example of a pillar being manifested at the system level, consider learning. One might suggest that learning rate naturally applies to individual species, but in our opinion it is more effectively interpreted at the system level. This is because every species evolves in concert with other species and its abiotic environment. Hence, when one species learns, other species in the system that do not become extinct must also learn. Even though humans are doing a great deal of learning, other species that we share the planet with are learning how to cope with the consequences and changes due to our learning (e.g., fungal species learning to degrade plastics [48,49,50] or humans learning to counter pathogens with antiobiotics, which then learn resistance strategies [51]).

We note also that the importance and relevance of the pillars are separate from the ease with which we can measure their presence. For example, the measurement of the autocatalytic property is likely to be nontrivial, unless the organisms/system in question can be cultured. In particular, biosignatures from distant exoplanets will likely not be sufficient to reveal the autocatalytic property, unless those signatures are collected over very long periods. Hence, for this pillar there is a difference between significance (this property is fundamental to the definition of lyfe) and ease of measurement (it may not be easy to assess).

However, sometimes the autocatalytic property reveals itself in dramatic fashion. As we write, the world as we know it is being turned upside down by an entity that is both considered not to be alive and comprises only a small RNA genome and a small set of proteins. Every chart that has tracked COVID-19 has shown exponential growth (autocatalysis) in the first phase, demonstrating how tiny biological entities can show extreme nonlinear dynamical changes in a short time.

### 2.1. Sublyfe

Beyond granting a simple “checklist” of criteria for determining whether or not a dynamical system is alyve, the four pillars also help us place lyfe in the context of other phenomena in the universe. We define lyfe as any system that performs all four pillars and sublyfe as any system that performs some but not all of those functions (see Figure 5).

Below, we list examples of phenomena that correspond to the numbered regions in Figure 5.

**Dissipation only:** Thermal diffusion, or any thermodynamically irreversible process.**Homeostasis only:** An ideal gas at equilibrium. An isolated system such as this always relaxes back to equilibrium after an internal or external fluctuation.**Dissipation and autocatalysis:** Fire is a frequently discussed example of dissipation and autocatalysis. It exhibits homeostasis of certain variables (e.g., burn temperature naturally stays within certain bounds), but its inability to fully regulate its behavior or learn from experience keeps it relegated to the nonliving world. Another relevant example would by the exponential growth of products in nonlinear chemical reactions (e.g., the formose reaction).**Dissipation and homeostasis:** A damped harmonic oscillator converts kinetic energy to thermal energy and always returns to its equilibrium position.**Dissipation and learning:** An artificial neural network is an example system that learns and is dissipative but does not necessarily exhibit autocatalytic growth or homeostasis (e.g., it does not by itself maintain the temperature of its own hardware). One could argue that their usefulness compels us to produce them at an exponential rate, but that is another discussion.**Dissipation, autocatalysis, and learning:** A living system that wipes itself out by tragedy of the commons. Examples might include invasive species introduced to an island that destroy their food sources so fast that the food sources are damaged beyond recovery. One might also suggest anthropic climate change as another example. Note that these cases depend critically on where one draws the boundary of the system (e.g., to include humans or not). Indeed, this form of sublyfe or sublife is less likely to occur because if the system is capable of learning, then in principle it could learn how to regulate itself homeostatically (unless it cannot learn fast enough).**Dissipation, homeostasis, and learning:** A “smart” house thermostat that monitors occupant behavior over time. This system cannot replicate but consumes free energy, is capable of primitive learning, and can regulate its local temperature.**Dissipation, autocatalysis, and homeostasis:** Thermal Gray–Scott reaction–diffusion spots. Certain nonequilibrium chemical patterns have been shown to grow exponentially and also regulate their local temperature [52,53,54,55].**All four:** Lyfe (which includes life).

Regarding homeostasis in equilibrium systems (region 2 in Figure 5), there are some delicate subtleties. Our contention that homeostasis occurs in such isolated systems is simply the fact that they are the archetype of stability (by definition). However, the point is arguable from both sides. One might argue that at the instant a fluctuation occurs, there is a momentary creation of free energy. However, harnessing such a fluctuation would require measurement and information-processing, and, as [56] showed, to accomplish this in a finite memory system requires erasure, which cannot be done for free and hence there is in fact no free energy created by fluctuations of an equilibrium system. In general, any perturbation of such a system will disappear in finite time as the system re-relaxes, hence we allow for homeostasis in equilibrium systems (in contrast, biological homeostatic processes generally occur through the consumption of free energy).

There are several notable macromolecular complexes that are both dissipative and autocatalytic but not necessarily homeostatic (regions 3 and 6 of Figure 5). We already discussed viruses above, which are protein–gene complexes. Amyloids (prions) are essentially peptide conformational viruses, since they propagate via a pathological spread of their own conformation on existing peptides (of different conformation) [57,58,59]. There are also autocatalytic genes, known as transposons: “transposable or mobile elements capable of parasite-like proliferation in the host genome” [60,61,62,63]. If we also consider the biochemical examples of self-replicating micelles and droplets [64,65], we see that the subset of entities that are dissipative and autocatalytic potentially deserve a dedicated category of their own. We note that such a category would also include phenomena at higher levels in the hierarchy of life. For instance, Internet memes are clearly autocatalytic and dissipative (given the computational energetic costs involved in their proliferation and communication). Furthermore, the social-media-membership-plus-meme system learns collectively, placing it in region 6 of Figure 5. Furthermore, in analogy to biological viruses, misinformation stories that are only weakly correlated with reality can be amplified by various effects to the point of causing disruptive socio-political impacts.

In this subsection, we have presented a list of increasingly lyfe-like phenomena, which may give the illusion that the origins of lyfe always proceed in a simple, stepwise manner—i.e., that a prebiotic dissipative structure must first exhibit exponential growth (perhaps replication), acquire homeostatic regulatory abilities, then finally learning. In our view, it is also plausible that relatively simple systems capable of rudimentary information processing can arise de novo and that the ability of these systems to optimize over time their dissipative, autocatalytic, and homeostatic traits will determine their ultimate fate (see also [66]). For instance, the first life on Earth almost certainly did not use DNA for information storage or any recognizable enzymes in its metabolic network. After innumerable tugs of war between chance and necessity, evolution produced the familiar macromolecules that we observe today. In Section 2.2, we turn to how the concept of lyfe can change our approaches to investigations of the origins of life.

### 2.2. Lyfe and Origins-of-Life Studies

With regard to the origins-of-life narratives discussed in Section 1.1, lyfe encompasses any system that fulfills the four pillars described above but may fulfill the three classic privileged functions—replication, metabolism, and compartmentalization—using components not used by earthly life. The idea that these three privileged functions define the necessary and sufficient conditions for life was explored in great theoretical detail within the artificial life field and the concepts of autopoiesis [67] and the “Chemoton” [68].

Figure 6 shows a “cube” whose vertices represent living systems with different combinations of components fulfilling the three privileged functions. At one vertex, life fulfills these privileged functions using the mechanisms of RNA/DNA, chemiosmosis, and lipid membranes. Directly connected to life are instances of lyfe that share two of the three mechanisms with life. Two steps removed from life are instances of lyfe that share only one of the three mechanisms with life. Furthermore, at the opposite vertex is a lyfe form completely unlike earthly life.

In Figure 6, life is positioned at the apex, because this is the end goal of most origins-of-life hypotheses. In origins-of-life narratives that involve stepwise emergence, an abiotic system might follow a trajectory that moves from one vertex to another, finally arriving at life. An example of this is illustrated by the dashed purple line in Figure 6, which denotes a narrative in which the abiotic synthesis of RNA was the first step in life’s onset, followed by compartmentalization inside of lipid membranes, followed by the emergence of chemiosmotic metabolism (e.g., an RNA world hypothesis). The way this diagram is drawn, origins-of-life narratives would be described by trajectories that move “upward” towards a convergence upon the general biochemical composition of life as we know it.

However, the “cube” shape of Figure 6 allows one to easily rotate the figure in the mind’s eye so that any combination of fulfilling mechanisms to the three privileged functions is at the apex—perhaps as a lyfeform would see it. Performing this mental gymnastics sparks two realizations: (1) The earliest stages of life’s evolution may have passed through phases that fully meet the living criteria but that we would identify as lyfe; (2) Origins-of-life research should be open to imagining and seeking lyfe-like solutions to the three privileged functions because they might represent attractors that our particular biological history did not find or settle in but alien lyfe did. The more detailed reasons for why a system might traverse these alternative pathways or not are a vast topic of philosophical debate, beyond the scope of the present work.

As discussed in Section 1.2, it is difficult to discern the components that the first life form used based on the highly evolved and complex machinery that present-day life uses. In the spirit of metaphor, we would like to highlight two analogies that are helpful when considering these issues. The first is the striking differences in composition and structure between scaffolding and finished products. The second is an analogy between biological evolution and the history of locomotives (Figure 7).

Consider a state-of-the-art Japanese Shinkansen (with a “metabolism” run by superconducting magnets cooled with liquid helium, “compartmentalization” made of composite materials such as carbon fiber, polymers and alloys, and “information processing” by a computer guidance system). By examining the Shinkansen and knowing nothing else, could one deduce the origin of the locomotive (a “metabolism” run by burning fossil fuels to boil water into steam, “compartmentalization” made of cast iron and wood, and “information processing” in the form of a human conductor)?

Given that train components can be swapped wholesale for one another by human hands, this analogy may appear inappropriate with regard to the commonly held view of biological evolution as a gradual process that is driven by the natural selection of minute tweaks to pre-existing elements. For example, in so-called “onion hypotheses” regarding the evolution of life [69,70,71], it is assumed that life began with a metabolic core of reactions, around which all further layers of complexity were laid.

However, as [72] argued, the evolution of life has progressed through several “major transitions,” many of which reshaped biology’s information processing capabilities. It is quite possible that at the earliest stages of life, such transitions were rampant, as primitive living systems battled for supremacy in a relatively flat fitness landscape, trading components with ease via horizontal transfers of information and material. Upon the invention of a mechanism that conferred greater selective advantage than the rest, that machinery—be it the ribosome or ATP synthase—would become a stable attractor in earthy biology’s evolutionary trajectory.

In his book *Seven Clues to the Origin of Life*, ref. [73] analogizes the lineage of life to a long rope made of overlapping fibers, in which no one fiber stretches from beginning to end. He writes: “There is then a simple way in which the central control machinery of organisms could have been updated: through a gradual takeover. A rope of hemp fibers at one end could gradually transform into a rope with only sisal fibers in it, by hemp fibers fading out and sisal fibers fading in.”

In our opinion, the truth is probably closest to a hybrid of these rope and onion concepts. Life’s emergence and early stages are probably well represented by the rope concept, with different fibers joining, contributing, and leaving the rope. Building from the “long rope” analogy, we suggest that Cairn-Smith’s “takeovers” could have been sudden rather than gradual.

However, once evolution stumbled upon a fiber with superior efficiency or stability, that feature would rapidly become integrated and grow into a central thread of the rope of life, forever ingrained in the biosphere’s molecular core. Layers of function accumulating around the rope core would prevent core components from being lost from the rope. This results in an “onion-like” history that can be peeled apart via modern techniques like molecular phylogeny and metabolomics.

Note that the ‘locking-in’ of functionality occurs in many complex systems including economic, technological, and political systems. Core systems that interface with many higher subsystems become so functionally entwined with the various modules that any changes to the core systems would be detrimental or catastrophic to the coherent whole. This can be true even if there are superior candidate systems for the core. For example, many European cities grew organically over long periods and now have a somewhat fractal organization. Compare this to the much more planned grid structures of American cities. Changing the fundamental structure of European cities is now too costly so they are stuck with their undesigned layouts.

It is also quite plausible that the modern components of core biochemical systems performed different functions originally. The biosphere is replete with examples of such exaptation: the shift in function or co-optation of a component for another use. Feathers originally granted warmth and signaling abilities before they became tools of flight; wings were likely used to augment running speed before being used for flight [74]; the water oxidizing complex that ushered in the age of oxygenic photosynthesis may have originally been used for manganese oxidation [75]. For further examples of exaptation, see the seminal work by [76] and citations therein.

Thus, we should be careful not to unequivocally equate a certain component’s universality in the modern biosphere with its necessity at the emergence of life. Perhaps long strands of nucleotides were not the first information processing system; perhaps polyphosphate chains were not the first “energy currency” of life; perhaps the first membranes were not composed of organic hydrocarbons. Origins-of-life narratives that seek abiotic pathways to synthesize the specific “building blocks” of life operate under the assumptions that those molecules: (1) were involved in the origins of life; (2) performed the same functions at the origins of life that they do today; (3) exhibited functionality that follows form, i.e., once all of the materials of life were synthesized and colocated, the complex information processing abilities of life would simply emerge (some have even described this third assumption as a form of modern-day “vitalism” [77]). This narrow approach is necessarily blind to scenarios where life-likeness could begin using alternative components, as might have occurred not only on Earth but elsewhere in the universe.

So, when investigating the emergence of life, what are we looking for if not specific biomolecules? The definition of lyfe offers an answer: we seek a system that exhibits dissipation, autocatalysis, homeostasis, and learning. By basing the criteria for lyfe on generic processes—rather than specific components that perform specific tasks—we open our minds to the exploration of all systems that display these emergent properties, freeing ourselves from the restrictions of precise chemical recipes whose prescriptions contain assumptions that may limit our explorations of the emergence of life-like behavior in the universe.

## 3. Imagining Lyfe

### 3.1. Examples of Alternative Components in Origins-of-Life Hypotheses

The origins-of-life community has suggested many hypotheses that involve stepwise emergence through lyfe-like phases. Using our classification, these emergent systems could be classified as lyfe or sublyfe because they utilize radically different components than present-day life to achieve one or more of the four pillars of lyfe. This section describes a few salient examples of origins hypotheses that utilize alternative components at the origin of life.

Some authors have hypothesized that the earliest metabolic systems may have been thioester-driven [70] (also note the concerns raised by [78]), rather than phosphate-driven, due to the inaccessibility of inorganic phosphate on the early Earth [79] (also note contradictory opinions [80]). Phosphorous is a key constituent of modern biology; it is found in ATP, the universal energy currency of life, metabolic cofactors like NADH, and information storage molecules like DNA and RNA. However, ref. [81] identified a plausible phosphate-free metabolism using a systems biology approach. Within the biosphere, a phosphate-independent metabolism exists that heavily depends on iron-sulfur and transition metal enzymes—which themselves have been linked to geochemical scenarios for the emergence of biochemistry [82,83,84]. Network-based algorithms suggest that various environmental parameters could result in a thioester-based proto-metabolic network similar to the reductive tricarboxylic acid cycle [85]. This would result in a “thioester world”, which, once phosphate became bioavailable, transitioned to the modern “phosphate world” that features ATP and nucleotides [70]. Thus, the thioester world, if it ever existed, would have been a primitive but thriving biosphere of lyfeforms.

Self-replicating clay minerals have been suggested as the first informational structures of life [86,87]. In this hypothesis, information is contained in the defects, irregularities, and aperiodic features in crystals’ chemical and spatial structures. Because crystals grow through the addition of planar, complementary layers, this information can be “replicated” when growing crystals break. These crystal genotypes can manifest as phenotypes that affect the environment in which the crystal is growing as well as the catalytic power of the crystal [73]. Of particular interest is the autocatalytic potential of clay–organic systems. For example, it has been suggested that the origin of life could have involved iron-rich clays that could perform light-induced charge transfer to reduce CO${}_{2}$ to functional organic molecules [69]. Clay minerals are also known to promote polynucleotide synthesis [88,89]. In this hypothesis for the emergence of life, an “organic takeover” would eventually ensue, in which organic–organic autocatalysis superseded clay–organic systems. Thus, this scientific narrative has clay minerals providing the scaffold for organic biochemistry: a clay–organic form of lyfe.

Recently, green rust (formally known as fougerite), an Fe${}^{2+}$/Fe${}^{3+}$ oxyhydroxide, has been suggested as the fundamental seed of life [90]. This metastable mineral is hypothesized to have been a constituent of Hadean submarine alkaline hydrothermal vents. Due to its reduction-oxidation capabilities, green rust is thought to have been an important driver of organosynthesis in these contexts [90]. Within its flexible, anhydrous interlayers, nitrate can be converted to ammonium [91,92], and it is hypothesized that inorganic carbon can be converted to pyruvate, which can then be aminated to alanine. At the same time, the theory suggests that green rust could act as a primitive inorganic pyrophosphatase—making it one of the first nanoengines of life [93]. Some have even hypothesized that the variable Fe${}^{2+}$/Fe${}^{3+}$ cations might have served as a primitive information-storage system [86,93,94]. If green rust (or a similar mineral) played the role of a molecular machine at the emergence of life, then that protometabolic system could be classified as lyfe.

In emergence hypotheses that occur in hydrothermal systems, compartmentalization is not achieved through spontaneously assembling lipid membranes but by a maze of inorganic mineral pore spaces. Only after the invention of lipid biosynthesis and the subsequent invention of cell wall biochemistry did life-like cells emerge and escape their hydrothermal confines [95]. However, before they became free-living prokaryotes, the complex chemical networks lodged inside their mineral enclosures could still be considered alyve (or subalyve before achieving all four pillars of lyfe).

Other researchers have considered amyloids—polypeptides with a unique β-sheet fold—as the first self-replicating biological entity. It is a longstanding problem that RNA synthesis and polymerization is a challenge in most abiotic environments (e.g., [96,97]), although several advances have been made in this area recently (e.g., [98]). Amyloid-based replicators offer an alternative to RNA: their monomers are more readily synthesized in prebiotic scenarios; they are stable under early Earth conditions; they are self-assembling, replicative, catalytic, and may be capable of adapting to changes in their environment [57,58]. Although amyloidosis is most famously known for being associated with human diseases [99], modern life uses functional amyloidal proteins to its benefit, from biofilm formation to dehydration resistance to long-term memory [58]. Should the earliest living entities have comprised an “amyloid world,” rather than an “RNA world,” this would represent a unique type of lyfe.

### 3.2. Lyfe on Titan

The biochemistry of life—from its water-based nature to its heavy reliance on CHNOPS-is a reflection of Earth’s physical and chemical conditions. Other worlds that feature potentially similar habitats—for instance, hydrothermal systems on Europa or Enceladus—may be inhabited by living creatures that are biochemically similar to life. However, on worlds that occupy radically different physical and chemical spaces to those of Earth, any exobiology that exists would surely be lyfe, not life. One intriguing possibility is Titan, the only other world in the Solar System known to possess stable bodies of liquid on its surface. Despite Titan’s broad-brush similarities to Earth (atmospheric pressure, geomorphology, an active methalogical cycle akin to Earth’s hydrological cycle, etc.), if exobiology exists on Titan, it would have emerged and evolved in an environment incredibly dissimilar to Earth. Major environmental differences include: extremely low surface temperatures of ∼94 K (resulting in exotic Van der Waals-driven organic chemistry [100,101]); a lack of available oxygen for biochemistry (resulting in hypotheses for N-substituted biochemistries [102]); and a nonpolar CH${}_{4}$-C${}_{2}$H${}_{6}$ solvent (resulting in speculations for alternative membrane structures).

Addressing the final difference in more detail, ref. [103] theorized that acrylonitrile (C${}_{2}$H${}_{3}$CN) could form stable membrane-like structures called azotosomes in Titan’s lakes and seas. Although acrylonitrile has recently been detected in Titan’s atmosphere [104], recent quantum mechanical calculations have shown that azotosome membranes do not self-assemble in Titan-like conditions [105]. However, this may not rule out the possibility that lyfe-like entities can construct them. Alternatively, membranes may not be required for lyfe on Titan because the extraordinarily low temperatures already prevent the dissolution of macromolecules, and membrane-like structures would inhibit the diffusion of metabolites in and out of a cold, stationary lyving system [105]. Should such a membrane-less exobiological entity exist, it would certainly be a provocative example of lyfe.

### 3.3. Mechanotrophs

In the spirit of imagining lyfe as we do not yet know it, we now present a hypothetical form of life that uses an alternative dissipative/metabolic system: the transduction of mechanical work into chemical disequilibria. Earthly life utilizes various external free energy sources, from redox couples to solar radiation to gradients of proton density and even electron density (e.g., electrotrophs [106]). Macroscopic organisms are known to exploit abiotic sources of mechanical work and use them to their advantage; examples of this include salmon hitching rides on turbulent eddies to swim upstream [107,108], birds catching thermals to higher altitudes, and humans building wind turbines and hydroelectric power plants. However, there is no cataloged example of an organism that transduces mechanical work directly into its metabolism.

We find this rather surprising, since the reverse process is such a fundamental component of all life, i.e., the conversion of chemiosmotic gradients to rotational motion using the ATP synthase family of molecular motors [109]. This rotational motion is most well known for synthesizing ATP from ADP and P${}_{i}$, but it is also used to produce cellular movement in fluid environments. This occurs through the flagellar motor proteins, close relatives of the ATP synthases [110,111]. Flagellar motors dissipate the free energy of ATP hydrolysis in eukaryotes, or ion motive forces in prokaryotes, for the production of rotational motion. Rotation of the central rotor causes the flagellar filament to spin, and the structure of the filament allows the rotational motion to produce translational movement of the organism in water (this motion has been compared to the mechanism by which corks are thrust from a bottle by the rotation of a corkscrew [112]).

Why could this process not be reversed to power a mechanotrophic organism? Imagine a single-celled organism in flowing water that is anchored to a rock via a pilus-like tether filament (like a person parasailing behind a boat). Assume that such an organism is equipped with the type of flagellar motor proteins that have been studied in swimming bacteria or single-cell eukaryotes. The flow of water causes rotational motion of flagellar filaments, and this rotational motion is used for ATP synthesis (or perhaps some other endergonic chemical step in more exotic lyfe), as shown in Figure 8. Whether this is feasible depends on the thermodynamic and fluid dynamic conditions.

Note that such a mechanotrophic organism might have a different design to the archetypal swimming bacterium. For example, it may use a torsionally stiff tether attached to a molecular generator molecule (the flagellum being driven), and the whole cell might rotate around the tether-generator complex. In this case the flagellum would be fixed relative to the cell body (at the opposite end to the tether-generator complex) and would rotate with the cell. The hydrodynamic details, efficiencies, and energetics of such an organism will be explored in detail in a future work. Here we simply wish to highlight the possibility of such an organism. This would be a prime example of a lyfeform, though it is also possible that such an organism exists on Earth but has simply gone unnoticed to date.

## 4. Conclusions

We have presented a new conceptual definition of life that encompasses four pillars: dissipation, autocatalysis, homeostasis, and learning. Any physical system exhibiting all of these necessary and sufficient features could be classified as lyfe, our designation for the set of all systems fulfilling the criteria, including the more specific case of life on Earth. This is in contrast to the idea of privileged functions, the pursuit of which has guided much of the research in the origins-of-life field. Systems that show differing levels of similarity to life (e.g., a different metabolic system or information-processing apparatus), could be considered lyfe or sublyfe. This would include phases of life’s emergence before it arrived upon the familiar metabolic cycles and pathways, DNA/RNA information storage and processing, and lipid membranes.

In the context of astrobiology, our conceptual framework can help clarify the objectives and discussions surrounding origins and biosignature research. When seeking to explain the emergence of biology on Earth or signatures of exobiology that are directly analogous to life on Earth, it would be appropriate to refer to life. On the other hand, when addressing questions related to more general systems that share the properties of life but differ in chemical or structural specifics, it would be appropriate to refer to lyfe. For example, finding unicellular chemotrophs resembling archaea on Europa would constitute a discovery of life. On the other hand, finding methane-based biology on Titan would constitute a discovery of lyfe.

Whether exobiology can emerge from a multitude of abiotic states or just one is a profound scientific question. Indeed, our ignorance in this matter is what motivates a conceptual definition of life that is agnostic to specific components and highlights processes instead. Only by seeking other instances of lyfe can we know if life is a proper subset of lyfe or if life and lyfe are identical sets. By framing life as a subset of a larger class of hypothetical phenomena called lyfe, we encourage the search for lyfe as we do not know it. This search can lead us to a deeper scientific understanding of what, how, and why we are: If we find that lyfe exists only in the form of life, that would suggest that the specific molecular components of life dictate the functionality of the lyving state (in line with a reductionist point of view). However, if we find that lyfe exists in myriad forms, that would suggest that “lyfeness” is emergent.

Finally, we also suggest a hypothetical lyfeform that transduces mechanical energy from a flowing fluid into its metabolism, a so-called mechanotroph. Such an organism has not been cataloged, and hence remains in the realm of lyfe. However, since no one is looking for it, it may be hiding in plain sight on Earth, which would render it a member of the life set.

In closing, we facetiously submit that this journal should henceforth be rebranded as *Lyfe* because of its expansive scope and open-mindedness towards publishing ideas such as ours. 

## Figures and Tables

**Figure 1 life-10-00042-f001:**
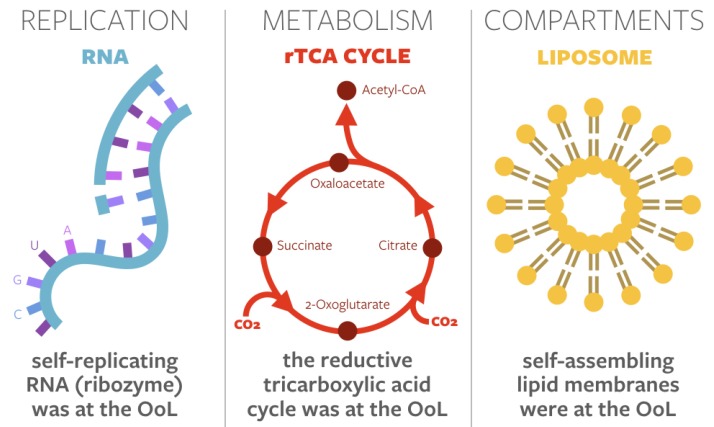
Three common examples of privileged functions in origins-of-life theories. After [2].

**Figure 2 life-10-00042-f002:**
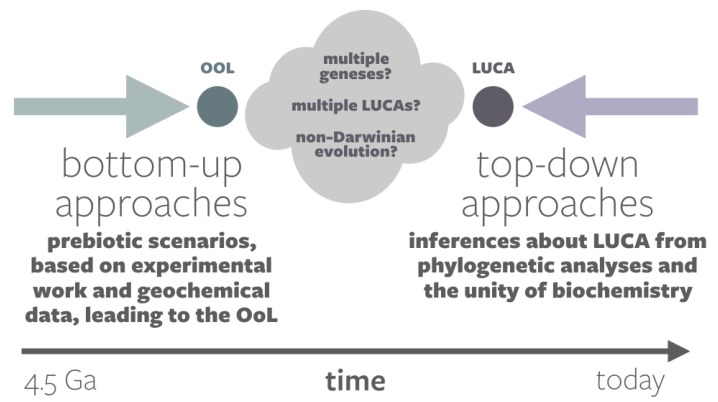
The event horizon in origins-of-life studies, denoted by a cloud of questions between bottom-up approaches, which seek to identify pathways to the origin of life, and top-down approaches, which seek to interpret the characteristics of Last Universal Common Ancestor(LUCA).

**Figure 3 life-10-00042-f003:**
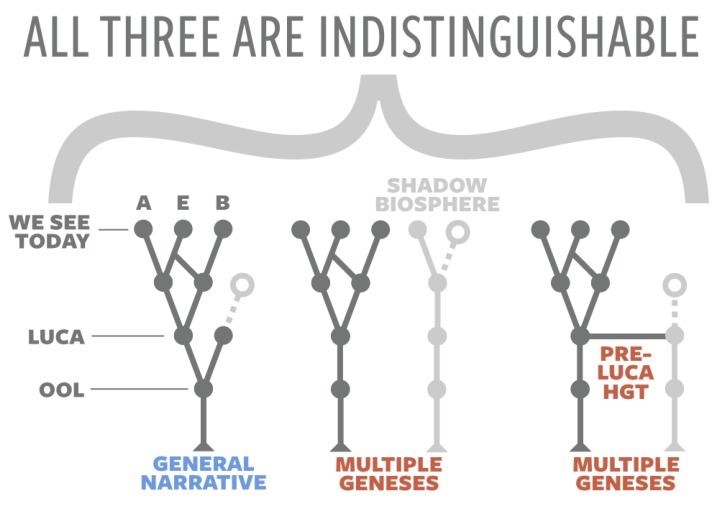
Top-down approaches cannot distinguish between the general narrative of a single origin of life on Earth and alternative narratives that involve multiple geneses resulting in extinct lineages and/or a shadow biosphere or narratives that involve multiple LUCAs that swapped genes via horizontal gene transfer (HGT). A = Archaea; E = Eukarya; B = Bacteria. Based on [18].

**Figure 4 life-10-00042-f004:**
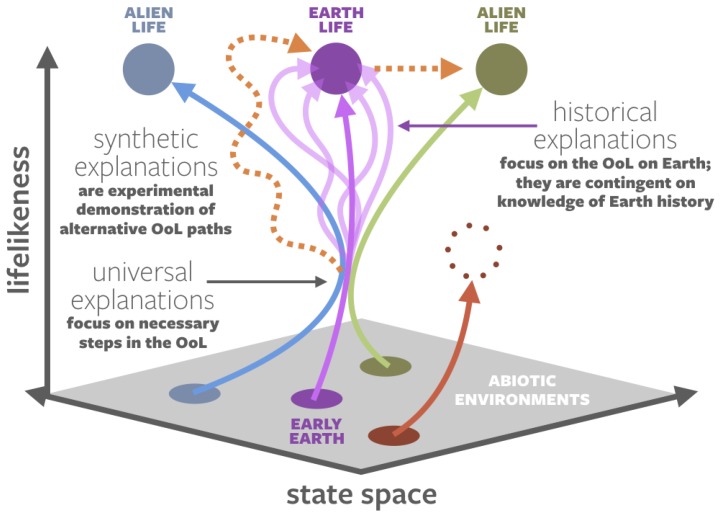
The three general categories of origins-of-life narratives: historical, synthetic, and universal. In purple are trajectories that describe historical narratives: how an abiotic Earth developed into life as we know it. The many trajectories with the same beginning and end points represent the different proposed hypotheses for the origin of life on Earth. In dotted orange are synthetic explanations. These may seek: (1) to recreate natural life on Earth, though likely through a different trajectory than the natural origin; (2) alter natural life to create new forms of life. In blue and green are hypothetical alien origins of life (lyfe); these and the trajectories that resulted in Earth life converge at a point in parameter space that universal narratives seek to describe. In red is a system that did not pass through those universal requirements, thus evolving into an end product that fulfills some but not all of life’s pillars. Based on [18,19].

**Figure 5 life-10-00042-f005:**
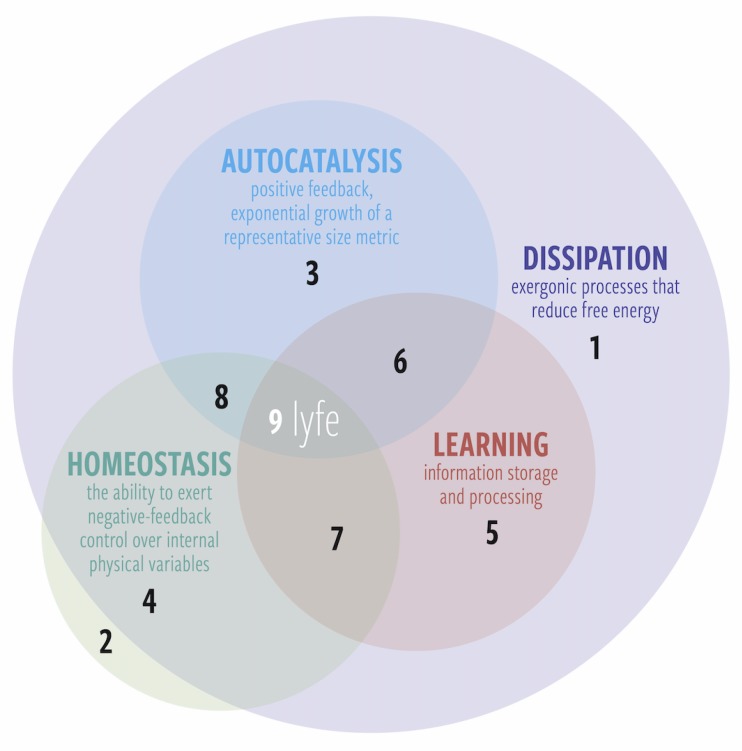
A Venn diagram of the four pillars of lyfe. Sublyfe (regions 1–8) is any system that performs some but not all of the pillars, while lyfe (region 9) is any system that performs all four. Autocatalysis and learning require a continuous supply of free energy and are thus contingent on dissipation; however, homeostasis can occur even in equilibrium systems and therefore does not always require dissipation. See text for examples of each region.

**Figure 6 life-10-00042-f006:**
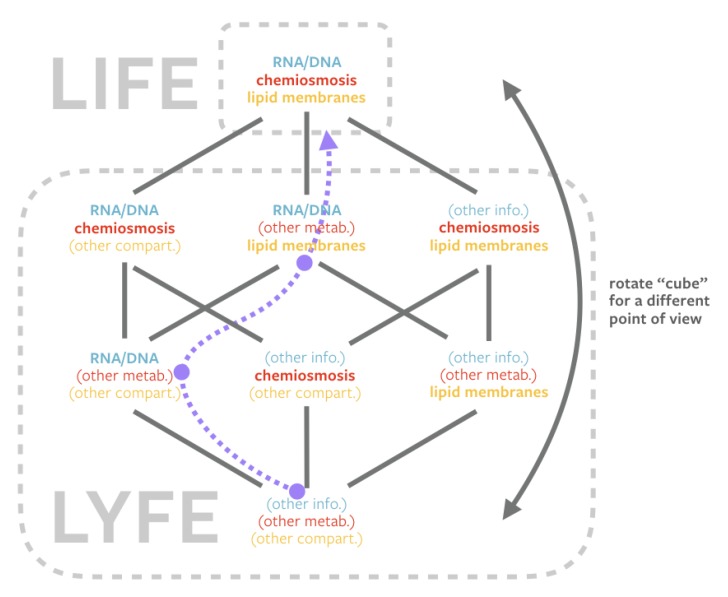
A “factor tree” of the three privileged functions: replication (blue), metabolism (red), compartmentalization (gold) (Section 1.1). Within the paradigm of privileged functions at the origins of life, life represents one solution to those functions in component-space, and lyfe encompasses any living system that uses other components to accomplish the same tasks. Life, being the sought-after end-product of origins-of-life hypotheses (one such shown by the dotted purple trajectory), is positioned at the apex. The tier directly below life contains two common components to life. The tier below that contains one common component. The final tier contains no common components. Connecting the vertices via common components creates a “cube”. Rotating this shape allows any combination to assume the apex, suggestive of our position that lyfe can be just as valid a target of origins-of-life research as life.

**Figure 7 life-10-00042-f007:**
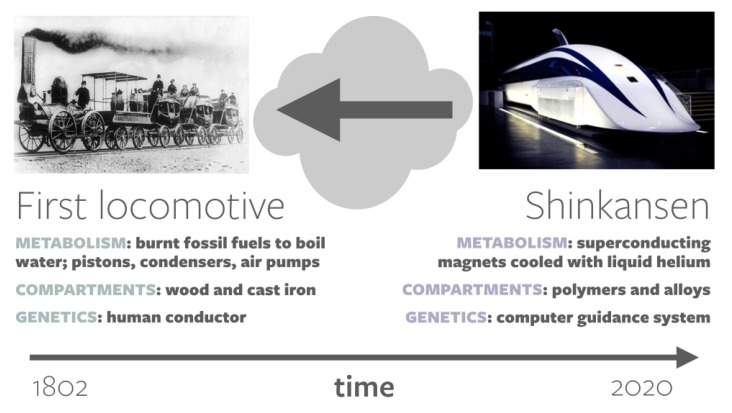
An analogy between the history of trains and biological evolution, presented in a form resembling Figure 2. Due to the many major transitions in train development, it would be a challenge to examine a modern-day Shinkansen and deduce in a “top-down” manner the components of the first train. While the materials that trains are built from have changed over time, their purpose and functions have remained constant. By analogy, the biological components that life uses today may be different from the biological components at life’s emergence, but the processes—the four pillars of lyfe—have been conserved. Shinkansen image: Daylight9899 (Wikimedia Commons).

**Figure 8 life-10-00042-f008:**
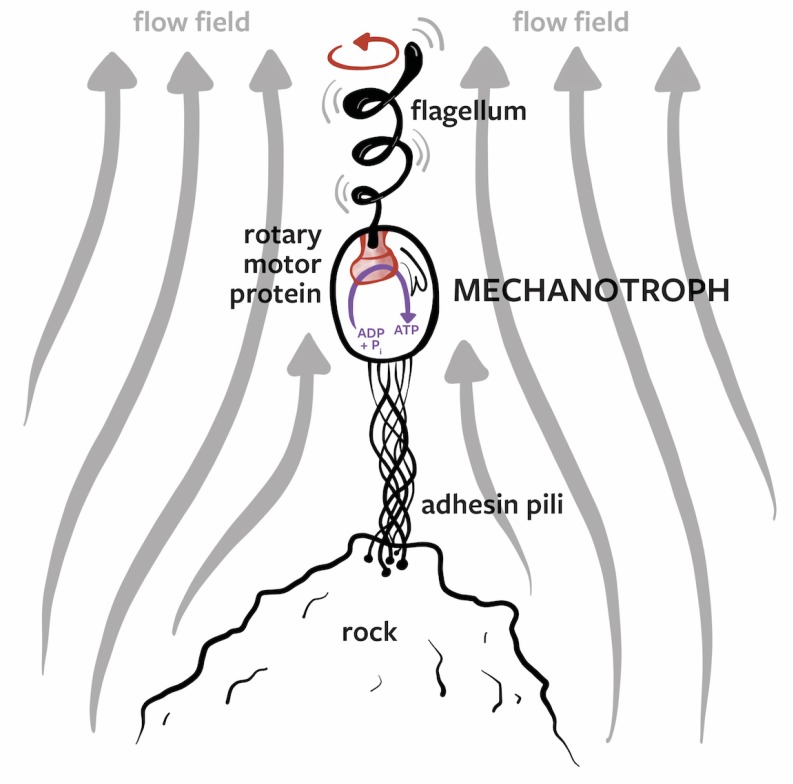
A hypothetical mechanotrophic lyfeform. This unicellular entity attaches itself to a rock via adhesin pili and transduces the mechanical motion of the surrounding fluid flow into intracellular free energy via the rotation of a motor protein (driven by the motion of a "flagellar turbine").

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
