# Peer review of "Defining Lyfe in the Universe: From Three Privileged Functions to Four Pillars"

_life, 2020, doi:10.3390/life10040042_

Round 1

Reviewer 1 Report

Comments on “Defining Lyfe in the Universe: From Three Privileged Functions to Four Pillars” by Bartlett & Wong

In this manuscript, the authors proposed an interesting term “lyfe” with four classifying standards. The goal is to solve our current controversies on the definition of life and broaden our criteria for life-search on other planets. Some thoughts are far-sighted and insightful.

The core of this study, i.e. “four pillars” of “lyfe” are “dissipation, autocatalysis, homeostasis, and learning”. I’m curious about the difference between “four pillars” and classical definition of life (homeostasis, organization, metabolism, growth, adaption, response to stimuli, reproduction). It appears that “four pillars” are summary of classical definition. Although the authors claimed that “four pillars” and “lyfe” are applicable to systems (which I agree on the importance of systems), it seems to me that some pillars, like “autocatalysis” (exponential growth in ideal conditions) are only applicable to species, not a system (like the Earth, Enceladus, or an island). Similarly, “learning” (which is called adaption in classical definition of life) is applicable to specific species (we cannot say the Earth or an island is learning, correct?).

With regards to autocatalysis, I also have another concern. Why "exponential growth" is necessary? In addition, the "ideal conditions" here is hard to define and vary largely among different systems. In reality, nearly all of systems are not in ideal conditions. How can you judge autocatalysis then?

Some specific comments:

  1. most of the references are cited only with year in main text. I think it is not correct format of citation and makes it hard to follow.
  2. line 229: with the example of island, how about microbes on the island? Although the food chain can be destroyed by invasive species (rabbit for example) and all plants and animals are dead, microbes are still alive, so it is still a living system.
  3. line 239: given “four pillars”, robot is lyfe here because under “ideal conditions”, it can reproduce “exponentially”. If it is armed with solar panel, it can maintain its only low-entropy state. It can also learn and maintain homeostasis. Similarly, other inorganic systems are probably classified as lyfe as well (provided “ideal conditions”). This makes the term lyfe very puzzling as the author seemed to focus on biochemical processes in all examples of lyfe but didn’t include any chemical constraints on lyfe (i.e. organic vs. inorganic).

Author Response

In this manuscript, the authors proposed an interesting term “lyfe” with four classifying standards. The goal is to solve our current controversies on the definition of life and broaden our criteria for life-search on other planets. Some thoughts are far-sighted and insightful.

We thank Reviewer 1 for their comments on our manuscript. We are grateful that the Reviewer found the concepts to be “far-sighted and insightful.”

The core of this study, i.e. “four pillars” of “lyfe” are “dissipation, autocatalysis, homeostasis, and learning”. I’m curious about the difference between “four pillars” and classical definition of life (homeostasis, organization, metabolism, growth, adaption, response to stimuli, reproduction). It appears that “four pillars” are summary of classical definition. 

We gratefully accept these comments. Indeed there is a degree of overlap between our four pillars and previous definitions of life. However, in our opinion, the four pillars offer the most concise and hopefully illuminating description of the living state. In the manuscript we tried to focus on the distinction between our definition and a) NASA’s standard definition, and b) the classical definition in terms of three privileged functions (metabolism, container and information). We acknowledge the interpretation of our definition as a condensation of the so-called classical definition. We see no issue with readers interpreting the work in this way, our primary objective is to clarify the physical and informational characteristics of life in general (lyfe). In our view the classical definition mentioned by the reviewer contains unnecessary and redundant components, e.g., there is considerable overlap between the concepts of ‘organization, ‘adaptation’ and ‘response to stimuli’. We hope that the four pillars contain the necessary and sufficient conditions for lyfe, where classical definitions sometimes contained more than the sufficient conditions (due to mutual conceptual redundancy).

Although the authors claimed that “four pillars” and “lyfe” are applicable to systems (which I agree on the importance of systems), it seems to me that some pillars, like “autocatalysis” (exponential growth in ideal conditions) are only applicable to species, not a system (like the Earth, Enceladus, or an island). Similarly, “learning” (which is called adaption in classical definition of life) is applicable to specific species (we cannot say the Earth or an island is learning, correct?).

Indeed the autocatalysis pillar can apply to subsystem characteristics such as number of individuals. Number of individuals is still a measure at the level of ecosystems however, if not higher. In general, the autocatalytic characteristic applies primarily to system-level properties. For example, human free energy consumption, a planetary-scale metric, has shown exponential growth in recent centuries. The rate of information processing of the entire planet has shown exponential-like growth over time as well. As the reviewer points out, learning rate naturally applies to individual species, but in our opinion it is more effectively interpreted at the system level. This is because every species evolves in concert with other species and its abiotic environment. Hence when one species learns, other species in the system that do not become extinct must also learn as well. So even though humans are doing a great deal of learning, other species that we share the planet with are learning about the consequences and changes due to our learning (e.g., fungal species learning to degrade plastics). We have added an extra paragraph before the beginning of section 2.1 to address this aspect.

With regards to autocatalysis, I also have another concern. Why "exponential growth" is necessary? 

In general, we see autocatalysis and exponential growth as two expressions of the same concept, i.e., when the presence/operation of a process increases the presence/operation of that same process. The natural consequence of any such positive feedback dynamic is exponential growth, due to the mathematical definition. 

In addition, the "ideal conditions" here is hard to define and vary largely among different systems. In reality, nearly all of systems are not in ideal conditions. How can you judge autocatalysis then?

Indeed ‘ideal conditions’ vary greatly depending on context. However the fossil record suggests that there were periods of evolutionary history where species abundances grew exponentially due to evolutionary innovation that led to the loss of an ecological constraint and proliferation into previously unexploited niches. 

We agree that the measurement of the autocatalytic property is likely to be non-trivial, unless the organisms/system in question can be cultured. In particular, biosignatures from distant exoplanets will likely not be sufficient to reveal the autocatalytic property. Hence, for this pillar there is a difference between significance (this pillar is fundamental to the definition), and ease of measurement (this pillar may not be easy to assess). However sometimes the autocatalytic property reveals itself in dramatic fashion. As we speak the world as we know it is being turned upside down by an entity that is both considered not to be alive and comprises a small RNA genome and a small set of proteins. Every chart that has tracked covid-19 has shown exponential growth (autocatalysis), demonstrating how tiny biological entities can show extreme nonlinear dynamical changes in short time. The above points have also been added in a new paragraph before the start of section 2.1.

Some specific comments:

  • most of the references are cited only with year in main text. I think it is not correct format of citation and makes it hard to follow.

We believe this is a formatting issue with the journal, as they reformatted our article before sending it out to the reviewers. We will make the appropriate changes and ensure that the citation style is correct before publishing.

  • line 229: with the example of island, how about microbes on the island? Although the food chain can be destroyed by invasive species (rabbit for example) and all plants and animals are dead, microbes are still alive, so it is still a living system.

This is a cogent point and we agree that the system itself could continue to live even if subsets of species have wiped themselves out. Indeed this example of sub-lyfe is less likely to occur because if the system is capable of learning then in principle it could learn how to regulate itself homeostatically (unless it cannot learn fast enough). These caveats have been added to Point 6 in Section 2.1.

  • line 239: given “four pillars”, robot is lyfe here because under “ideal conditions”, it can reproduce “exponentially”. If it is armed with solar panel, it can maintain its only low-entropy state. It can also learn and maintain homeostasis. Similarly, other inorganic systems are probably classified as lyfe as well (provided “ideal conditions”). This makes the term lyfe very puzzling as the author seemed to focus on biochemical processes in all examples of lyfe but didn’t include any chemical constraints on lyfe (i.e. organic vs. inorganic).

In the hypothetical scenario that the Reviewer describes, this robot would constitute a form of lyfe because it performs all four pillars of lyfe but is not composed of biochemical machinery. As the Reviewer has likely surmised, our definition of lyfe is inclusive of what is often called “artificial” or “synthetic” life. Because we specifically wish to divorce our definition of lyfe from the components of the system, we purposefully do not include any chemical constraints on lyfe. We focus on biochemical processes in the text because our thoughts on this matter were derived from origins-of-life studies.

Reviewer 2 Report

In the present article authors have proposed a new conceptual definition of life in a more expansive way while recognizing the need to signify the specific kind of life that earthly forms represent. The article thoroughly analyses the major issues needing a new definition of life and also examine how their definition of life also called lyfe might engender new perspectives on origin of life research. The article is scholarly and the illustrations are useful. The Venn diagram showing the four pillars of life is very much helpful in distinguishing lyfe from sub-lyfe. Overall, this review will immensely contribute to the origin of life communities with different schools of thought. I would recommend publication of the manuscript once following points are addressed.

Comment 1: Citation style in the text section needs to be corrected.

Comment 2: Clay minerals can increase the self-clevage of hammerhead ribozymes (Gene 2007, 389, 10) and can also mediate folding and regioselective interactions of RNA (J. Am. Chem. Soc. 2010, 132, 13750). These points can be discussed in the appropriate section of the paper.

Comment 3: The authors discussed about compartmentalization by lipid as one of the privileged functions of life. Compartmentalization is also beneficial for functional RNA molecules (Nat. Commun. 2018, 9, 2313 and Orig. Life Evol. Biosph. 2015, 44, 319) and worth mentioning.

Author Response

In the present article authors have proposed a new conceptual definition of life in a more expansive way while recognizing the need to signify the specific kind of life that earthly forms represent. The article thoroughly analyses the major issues needing a new definition of life and also examine how their definition of life also called lyfe might engender new perspectives on origin of life research. The article is scholarly and the illustrations are useful. The Venn diagram showing the four pillars of life is very much helpful in distinguishing lyfe from sub-lyfe. Overall, this review will immensely contribute to the origin of life communities with different schools of thought. I would recommend publication of the manuscript once following points are addressed.

We thank Reviewer 2 for their comments on our manuscript. We are grateful that the Reviewer found the manuscript to be “scholarly” and our illustrations “useful,” especially the Venn diagram (Figure 4). It was our goal to “contribute to the origin of life communities with different schools of thought,” and we are especially pleased that the Reviewer has judged our efforts to be successful in this endeavor.

Comment 1: Citation style in the text section needs to be corrected.

We believe this is a formatting issue with the journal, as they reformatted our article before sending it out to the reviewers. We will make the appropriate changes and ensure that the citation style is correct before publishing.

Comment 2: Clay minerals can increase the self-clevage of hammerhead ribozymes (Gene 2007, 389, 10) and can also mediate folding and regioselective interactions of RNA (J. Am. Chem. Soc.2010, 132, 13750). These points can be discussed in the appropriate section of the paper.

We thank the Reviewer for bringing these papers to our attention and have cited them in Section 3.1.

Comment 3: The authors discussed about compartmentalization by lipid as one of the privileged functions of life. Compartmentalization is also beneficial for functional RNA molecules (Nat. Commun. 2018, 9, 2313 and Orig. Life Evol. Biosph. 2015, 44, 319) and worth mentioning.

We thank the Reviewer for bringing these papers to our attention and have cited them in Section 1.1.

Reviewer 3 Report

The paper by Bartlett and Wong proposes a new, generalized definition of life (called “lyfe”) based on four main characteristics (autocatalysis, dissipation, homeostasis and learning) of the living state, as we know it from our planet.  First of all this is a bright and innovative concept, which is formulated in the paper in a clear (I would even say “entertaining”) manner.  The authors also introduce the term “sublyfe” for those systems that fulfill only part of the conditions necessary for lyfe.  Though I strongly believe in the privileged role of RNA at the origin of terrestrial life, I appreciate the viewpoint of the authors suggesting that the whole origin of life field would substantially benefit from considering other “lyfe-like” solutions instead of dogmatizing the importance of certain privileged functions at life’s emergence.  For this very reason I suggest publication of this fine work in its current form.

Author Response

The paper by Bartlett and Wong proposes a new, generalized definition of life (called “lyfe”) based on four main characteristics (autocatalysis, dissipation, homeostasis and learning) of the living state, as we know it from our planet.  First of all this is a bright and innovative concept, which is formulated in the paper in a clear (I would even say “entertaining”) manner. The authors also introduce the term “sublyfe” for those systems that fulfill only part of the conditions necessary for lyfe.  Though I strongly believe in the privileged role of RNA at the origin of terrestrial life, I appreciate the viewpoint of the authors suggesting that the whole origin of life field would substantially benefit from considering other “lyfe-like” solutions instead of dogmatizing the importance of certain privileged functions at life’s emergence.  For this very reason I suggest publication of this fine work in its current form.

We thank Reviewer 3 for their comments on our manuscript. We are grateful that the Reviewer found the concepts to be “bright and innovative” and delivered in a “clear” and “entertaining” manner. We appreciate their candor about their views on the privileged role of RNA, and we acknowledge their open-mindedness in considering our expansive definition of life worthwhile of publication.

Reviewer 4 Report

Bartlett and Wong addressed a common problem of the definition of life, a frequently discussed topic especially in the field of astrobiology and the origins of life. While I think many people acknowledge the problems raised by the authors, such that different researchers or parties are seeking different types of life or life-like systems, the authors move the step forward by defining “lyfe”, which harbors four key features related to living states. The authors first reviewed current issues about the definition of life, and then introduced the new term (lyfe) and the concept to help clarify what we should look for in the future research of astrobiology and the origins of life. The manuscript is on-target and well-organized; I personally enjoyed reading through it. I have just several comments that I would like the authors to address before the acceptance.

  1. L144-, etc.: The authors defined the term “lyfe” based on the fundamental processes of the living state of extant life (as we know it). I think lyfe is therefore constrained by life on Earth and does not necessarily include some other hypothetical life. I would like an explanation about why the authors think the definition of such lyfe is sufficiently universal.

  1. L175: I think the definition of autocatalysis need to be more specific. Does it include collective autocatalysis, such as cross catalysis? The example of autocatalysis in life described at the place is a simple autocatalysis, a direct positive feedback loop.

  1. L187: I suggest the authors describe the relationship between learning and evolution. The examples of learning in life in italic does not include evolution, but at L202, the authors indicated that learning can occur through evolution. I would argue, however, that learning and evolution may be different because according to the authors’ note, learning is an active process that records and treats information to carry out actions better, but I do not think (at least Darwinian) evolution is in a sense that mutations are introduced randomly and fitter ones that can carry out actions better are just selected (increase their relative frequencies) since they produce more progenies. (But I believe evolution is a fundamental process of living states.)

  1. L196-: As the authors noted, viruses could meet all the four pillars in a system level. Virus can therefore be considered at least “lyving” (depending on environments), which I think is reasonable because whether a virus is alive is still controversial. However, in that case, I think broader examples, including ones that no one may have thought alive, could be categorized in lyfe as well. For example, a recent artificial cell-like system described in (Mizuuchi & Ichihashi, 2018) seems to have shown all the four pillers: dissipation (such as NTP-NDP conversions to maintain genome replication), autocatalysis (a genome-encoded replication enzyme replicates the genome), some extent of homeostasis (the system is resilient to strong dilution; it can be back to the original replication dynamics immediately), and learning (Darwinian evolution, if it is categorized in learning), all of which worked probably only in a controlled laboratory environment with sufficient supply of materials, as viruses function only in a system level. Although I think this example is far simpler and much less natural than viruses, should it be treated as lyfe too? Is there any other examples of lyfe, yet certainly not life, created in a lab?

(Mizuuchi R, Ichihashi N. 2018, Sustainable replication and coevolution of cooperative RNAs in an artificial cell-like system. Nature Ecology & Evolution, 2: 1654–1660.)

  1. L229, I got confused here because I originally thought life cannot be non-lyfe, although non-life can be lyfe. However, the authors indicated that a living system (that I think is certainly life) can be non-lyfe depending on a situation, which I think contradicts with statements such as L4 (“Life is defined as the instance of lyfe…”).

  1. L465: This is an interesting question. The authors focused on single-celled organisms, but do the authors still find no examples of such species if the focus is moved out of single-celled organisms (some species habitat on a rock through temporal adhesion (e.g., shellfish))? As the authors noted, the existence of such a mechanotrophic organism surely depends on thermodynamics and fluid dynamics of habitats, but it seems to me that there are numerous possible places (on Earth) seemingly compatible with the unique lyfestyle.

Author Response

Bartlett and Wong addressed a common problem of the definition of life, a frequently discussed topic especially in the field of astrobiology and the origins of life. While I think many people acknowledge the problems raised by the authors, such that different researchers or parties are seeking different types of life or life-like systems, the authors move the step forward by defining “lyfe”, which harbors four key features related to living states. The authors first reviewed current issues about the definition of life, and then introduced the new term (lyfe) and the concept to help clarify what we should look for in the future research of astrobiology and the origins of life. The manuscript is on-target and well-organized; I personally enjoyed reading through it. I have just several comments that I would like the authors to address before the acceptance.

We thank Reviewer 4 for their comments on our manuscript. We are grateful that the Reviewer found the manuscript “on target and well-organized” and that they “personally enjoyed reading through it.” We concur with the Reviewer that the issues we raised in the Introduction have been acknowledged by other authors. We are grateful that the Reviewer agrees that our “step forward” in defining lyfe, motivated by a desire to ameliorate the issues, is a novel contribution worthy of publication.

  • L144-, etc.: The authors defined the term “lyfe” based on the fundamental processes of the living state of extant life (as we know it). I think lyfe is therefore constrained by life on Earth and does not necessarily include some other hypothetical life. I would like an explanation about why the authors think the definition of such lyfe is sufficiently universal.

We thank the Reviewer for bringing up this is a very astute observation. Indeed, it is something that we have also given some thought. While our definition of lyfe is “influenced” by life as we know it on Earth (this is inevitable), we do not believe that this contradicts its universality. We have elaborated on this issue in Section 2 by adding the following paragraph to our manuscript:

While these four pillars of lyfe are derived from observations of life as we know it (after all, life must be a subset of lyfe), this new definition is far more expansive. The four pillars constitute necessary and sufficient requirements of the lyving state while remaining divorced from the specific components that make up the system. This is illustrated by the fact that there are numerous systems that perform the same pillars but are quite distinct in form (as will be discussed in subsection 2.1). Hence, the universality of the term “lyfe” is derived from the reasonable expectation that it can be applied to as yet undiscovered (or uninvented) systems that exist at myriad scales across the universe. There may even be a class of systems, still undiscovered and undescribed, that performs all four pillars of lyfe and some fifth pillar as well. Such systems might be deemed “super-lyfe.” While the discovery of super-lyfe would certainly be paradigm-shifting, we remain agnostic about its existence for now.

  • L175: I think the definition of autocatalysis need to be more specific. Does it include collective autocatalysis, such as cross catalysis? The example of autocatalysis in life described at the place is a simple autocatalysis, a direct positive feedback loop.

We appreciate this cogent point. Our definition does include more general catalytic effects including cross-catalysis and network autocatalysis, as long as the effect leads to exponential growth of a suitable metric under ideal conditions. We accept that further work may be required to rigorously and unambiguously define this concept (hopefully to the point where all is defined in purely physical terms), but at this stage we simply wish to express the general principle. As an example that we have now mentioned just before the beginning of section 2.1, the novel coronavirus is tiny in terms of biomass and size, and yet it has exhibited autocatalytic and exponential growth in several different ways in a very short period. We have also added the suggested additions of cross catalysis and network autocatalysis in pt2 on page 7.

  • L187: I suggest the authors describe the relationship between learning and evolution. The examples of learning in life in italic does not include evolution, but at L202, the authors indicated that learning can occur through evolution. I would argue, however, that learning and evolution may be different because according to the authors’ note, learning is an active process that records and treats information to carry out actions better, but I do not think (at least Darwinian) evolution is in a sense that mutations are introduced randomly and fitter ones that can carry out actions better are just selected (increase their relative frequencies) since they produce more progenies. (But I believe evolution is a fundamental process of living states.)

We agree that this is a very important point. We also agree on the possible separation between the simplest form of evolution (heritable variation leading to reproductive success or Darwinism), and more subtle forms which may be more active (as opposed to simply waiting for random mutations to be selected for), and/or include recognizable learning processes, e.g., associative learning or Bayesian inference. Our position is that Darwinian evolution is a biological learning process among a much larger set of learning processes including neural learning, Lamarckian evolution, evolution of development, epigenetics, etc. In the biosphere, Darwinism mingles with these other learning processes and perhaps other hitherto undiscovered forms to create the incredible diversity and complexity of the biosphere. Hence we use learning as a generic umbrella term for this large and poorly understood set of functional information accumulation processes. This clarification has been added pt 4 on page 7.

  • L196-: As the authors noted, viruses could meet all the four pillars in a system level. Virus can therefore be considered at least “lyving” (depending on environments), which I think is reasonable because whether a virus is alive is still controversial. However, in that case, I think broader examples, including ones that no one may have thought alive, could be categorized in lyfe as well. For example, a recent artificial cell-like system described in (Mizuuchi & Ichihashi, 2018) seems to have shown all the four pillers: dissipation (such as NTP-NDP conversions to maintain genome replication), autocatalysis (a genome-encoded replication enzyme replicates the genome), some extent of homeostasis (the system is resilient to strong dilution; it can be back to the original replication dynamics immediately), and learning (Darwinian evolution, if it is categorized in learning), all of which worked probably only in a controlled laboratory environment with sufficient supply of materials, as viruses function only in a system level. Although I think this example is far simpler and much less natural than viruses, should it be treated as lyfe too? Is there any other examples of lyfe, yet certainly not life, created in a lab?

(Mizuuchi R, Ichihashi N. 2018, Sustainable replication and coevolution of cooperative RNAs in an artificial cell-like system. Nature Ecology & Evolution, 2: 1654–1660.)

We thank the Reviewer for bringing this intriguing study to our attention. The artificial cell-like entities created by Mizuuchi & Ichihashi (2018) are certainly fascinating systems, and the results of their experiments are very informative to the understanding of cooperation on the molecular scale. After careful review of Mizuuchi & Ichihashi’s paper, we hesitate to call their cell-like systems true lyfeforms because they existed solely in the realm of a particular experiment that relied heavily on human intervention (for the introduction of genetic mutations and the induction of droplet dilution, fusion, and division). Taking a systems-level view, we contend that it might be more accurate to call the system that includes the scientists, the experiment, and the cell-like entities a “lyving system.” We consider it quite likely that future laboratory experiments can yield novel forms of lyfe. Perhaps a system like Mizuuchi & Ichihashi’s cell-like entities that can fully self-replicate of their own accord and demonstrably learn about other environmental perturbations will be the first such case. At present this work stands alongside other attempts both in vitro and in silico to create a bona fide evolvable system. Attempts such as those by Craig Venter, Steen Rasmussen, Chris Adami and many others have so far failed to create a system that truly evolves open-endedly or solves the cooperation barrier problem. We do not deny the huge progress that has been made by these researchers but are not convinced that an artificial system is yet truly evolvable or can spontaneously learn (this problem is also analogous to the general intelligence quest in the field of artificial intelligence).

  • L229, I got confused here because I originally thought life cannot be non-lyfe, although non-life can be lyfe. However, the authors indicated that a living system (that I think is certainly life) can be non-lyfe depending on a situation, which I think contradicts with statements such as L4 (“Life is defined as the instance of lyfe…”).

If a living system does not maintain homeostasis in the long term then it fails to fulfil all four pillars and hence is a form of sub-lyfe. In the case of the island ecosystem or anthropic climate change, these are also sub-life, although if one draws the boundary around the system to exclude the collapsing species, then perhaps all four pillars will still be fulfilled after the collapse, by the remaining species. In general, this subset of lyfe is unlikely to occur in nature and we have now mentioned that in the text, along with an additional description of the issues described above. These caveats have been added to Point 6 in Section 2.1.

  • L465: This is an interesting question. The authors focused on single-celled organisms, but do the authors still find no examples of such species if the focus is moved out of single-celled organisms (some species habitat on a rock through temporal adhesion (e.g., shellfish))? As the authors noted, the existence of such a mechanotrophic organism surely depends on thermodynamics and fluid dynamics of habitats, but it seems to me that there are numerous possible places (on Earth) seemingly compatible with the unique lyfestyle.

As noted in the text, salmon “ratcheting” upstream using eddies and birds using thermals to loft themselves to higher altitudes are examples of macroscopic lifeforms (other than humans) utilizing flowfields to perform useful work. Our mechanotrophy concept envisions a lifeform that can couple mechanical energy into chemical/electrochemical disequilibria. Such a free energy transduction on the nanoscale has yet to be discovered on Earth. If a macroscopic lyfeform does employ mechanotrophy, we imagine it would likely do so by using an army of individual cells with flagellar motors on the periphery of the organism. The ancestral proteins involved in this task would have evolved in unicellular organisms, and all of the “action” would remain at the nanoscale. This is analogous to how macroscopic photosynthesizers still use nanoscale photopigments and reaction centers that emerged eons ago in cyanobacteria to transduce electromagnetic energy into chemical energy. Perhaps it is possible for macroscopic biological machinery to transduce mechanical energy into chemical energy, but this is difficult to imagine and we leave future researchers to ponder this question.